# An adenovirus-vectored COVID-19 vaccine confers protection from SARS-COV-2 challenge in rhesus macaques

Liqiang Feng[1,2,8], Qian Wang[1,3,8], Chao Shan [4,8], Chenchen Yang[5], Ying Feng[2], Jia Wu[4], Xiaolin Liu[5], Yiwu Zhou[6], Rendi Jiang[4], Peiyu Hu[1], Xinglong Liu[1], Fan Zhang[1], Pingchao Li[1], Xuefeng Niu[2], Yichu Liu[1], Xuehua Zheng[1], Jia Luo[1], Jing Sun[2], Yingying Gu[2], Bo Liu[5], Yongcun Xu[5], Chufang Li[2], Weiqi Pan[2], Jincun Zhao [2], Changwen Ke[7], Xinwen Chen[1], Tao Xu [1], Nanshan Zhong[2], Suhua Guan[5✉], Zhiming Yuan [4✉] & Ling Chen [1,2,5✉]

The rapid spread of coronavirus SARS-CoV-2 greatly threatens global public health but no prophylactic vaccine is available. Here, we report the generation of a replication-incompetent recombinant serotype 5 adenovirus, Ad5-S-nb2, carrying a codon-optimized gene encoding Spike protein (S). In mice and rhesus macaques, intramuscular injection with Ad5-S-nb2 elicits systemic S-specific antibody and cell-mediated immune (CMI) responses. Intranasal inoculation elicits both systemic and pulmonary antibody responses but weaker CMI response. At 30 days after a single vaccination with Ad5-S-nb2 either intramuscularly or intranasally, macaques are protected against SARS-CoV-2 challenge. A subsequent challenge reveals that macaques vaccinated with a 10-fold lower vaccine dosage ($1 \times 10^{10}$ viral particles) are also protected, demonstrating the effectiveness of Ad5-S-nb2 and the possibility of offering more vaccine dosages within a shorter timeframe. Thus, Ad5-S-nb2 is a promising candidate vaccine and warrants further clinical evaluation.

[1] Bioland Laboratory (GRMH-GDL), Guangzhou Institutes of Biomedicine and Health, Chinese Academy of Sciences, Guangzhou, China. [2] State Key Laboratory of Respiratory Disease, Guangzhou Institute of Respiratory Health, First Affiliated Hospital of Guangzhou Medical University, Guangzhou, China. [3] University of Chinese Academy of Sciences, Beijing, China. [4] State Key Laboratory of Virology, Key Laboratory of Special Pathogens and Biosafety, Wuhan Institute of Virology, Chinese Academy of Sciences, Wuhan, China. [5] Guangzhou nBiomed Ltd, Guangzhou, China. [6] Department of Forensic Medicine, Tongji Medical College of Huazhong University of Science and Technology, Wuhan, Hubei, China. [7] Guangdong Provincial Center for Disease Control and Prevention, Guangzhou, China. [9] These authors contributed equally: Liqiang Feng, Qian Wang, Chao Shan. ✉email: guan_suhua@gznbio.com; yzm@wh.iov.cn; chen_ling@gibh.ac.cn

Coronavirus disease 2019 (COVID-19), caused by the emerging coronavirus SARS-CoV-2, has rapidly swept throughout the world[1,2]. As of July 22, 2020, there have been more than 14 million laboratory-confirmed COVID-19 patients and nearly 612,000 deaths[2]. The World Health Organization has declared COVID-19 a public health emergency of international concern. Common symptoms of COVID-19 are fever, cough and lymphocytopenia, and chest radiographic abnormality[3]. A proportion of patients recovering from COVID-19 keep shedding virus for days[4], and asymptomatic carriers may also transmit SARS-CoV-2[5,6], indicating a risk of a continuous and long-term pandemic. Highly effective antiviral drugs are not available. An effective vaccine is urgently needed.

SARS-CoV-2 is an enveloped, single-stranded, positive-sense RNA virus belonging to the family *Coronavidae* and the genus β-coronavirus[7]. The genome of SARS-CoV-2 encodes one large Spike protein (S) that plays a pivotal role during the viral attachment and entry into host cells. Like other coronaviruses, the S protein can be cleaved into S1 and S2 subunits by host proteases. The S1 subunit contains the receptor-binding-domain (RBD) through which the virus binds to its receptor angiotensin-converting enzyme 2 (ACE2)[7,8]. The S2 subunit contains the fusion peptide and facilitates membrane fusion and viral entry[9]. The S protein has been frequently considered as the major antigen target for vaccines against human coronavirus such as SARS-CoV, MERS-CoV, and also SARS-CoV-2 in recent studies[10,11], because it contains the major epitopes targeted by neutralizing antibodies[9,12–14]. Neutralizing antibodies targeting the RBD may block viral binding to host cells, whereas those targeting the S2 subunit may inhibit membrane fusion and viral entry[9,12,15,16].

In this study, we generate a replication-incompetent recombinant adenovirus serotype 5 that carries a codon-optimized gene encoding the full-length SARS-CoV-2 S protein (Ad5-S-nb2). We investigate the antibody and cell-mediated immune (CMI) responses elicited by Ad5-S-nb2 via injection and non-injection routes in mice and rhesus macaques. We evaluate the protective efficacy of the candidate vaccine in rhesus macaques by an intratracheal challenge of SARS-CoV-2.

## Results

**Generation and characterization of Ad5-S-nb2.** In an attempt to develop a prophylactic vaccine against SARS-CoV-2, we constructed a replication-incompetent recombinant adenovirus, Ad5-S-nb2, that can efficiently express SARS-CoV-2 S protein in infected cells (Fig. 1a). The S-coding sequence was optimized by altering the codon usages to increase its expression in human cells. Compared to the original viral S coding sequence, transfection with a plasmid carrying the optimized S-coding sequence, S-nb2, enabled a significant elevation of S protein expression in human cells (Fig. 1b). We inserted the expression cassette containing human CMV promoter and S-nb2 into the E1 region of an E1 and E3 deleted Ad5 vector. Ad5-S-nb2 was successfully rescued and propagated in human embryonic kidney (HEK) 293 cells, which provide E1 products in trans to support the replication of Ad5-S-nb2. Infection of Ad5-S-nb2 in mammalian cells such as HEK293 cells and human lung carcinoma A549 cells resulted in efficient expression of S protein (Fig. 1c, d).

**Ad5-S-nb2 elicits antibody and CMI responses in BALB/c mice.** To assess the immunogenicity of Ad5-S-nb2, we first immunized eight-week-old female BALB/c mice. Group 1 and Group 2 received $1 \times 10^9$ and $5 \times 10^9$ viral particles (vp) per mouse via an intramuscular (IM) injection. Group 4 and Group 5 received $1 \times 10^9$ and $5 \times 10^9$ vp per mouse via an intranasal (IN) instillation (Fig. 2a). Group 3 and Group 6 that received $5 \times 10^9$ vp

Ad5-empty vector via an IM or IN route respectively were used as controls (Fig. 2a). An IM injection with $5 \times 10^9$ vp Ad5-S-nb2 elicited significant serum IgG responses against the S protein, RBD, and the S2 subunit as early as day 6 after vaccination, whereas $1 \times 10^9$ vp Ad5-S-nb2 had lower serum IgG titers against the S protein, RBD, and the S2 subunit, demonstrating a dose-dependent response (Supplementary Fig. 1a). The serum IgG responses against the S protein, RBD, and the S2 subunit continued to increase until day 28 when the mice were sacrificed for analysis (Supplementary Fig. 1a). On day 28 after vaccination, significantly higher serum IgG titers against the S protein, RBD, and the S2 subunit were observed in mice vaccinated with $5 \times 10^9$ vp Ad5-S-nb2 than in mice vaccinated with $1 \times 10^9$ vp Ad5-S-nb2 (Fig. 2b). An IN inoculation with Ad5-S-nb2 elicited a lower level of serum IgG responses against the S protein, RBD, and the S2 subunit on day 11 after vaccination (Supplementary Fig. 1b).

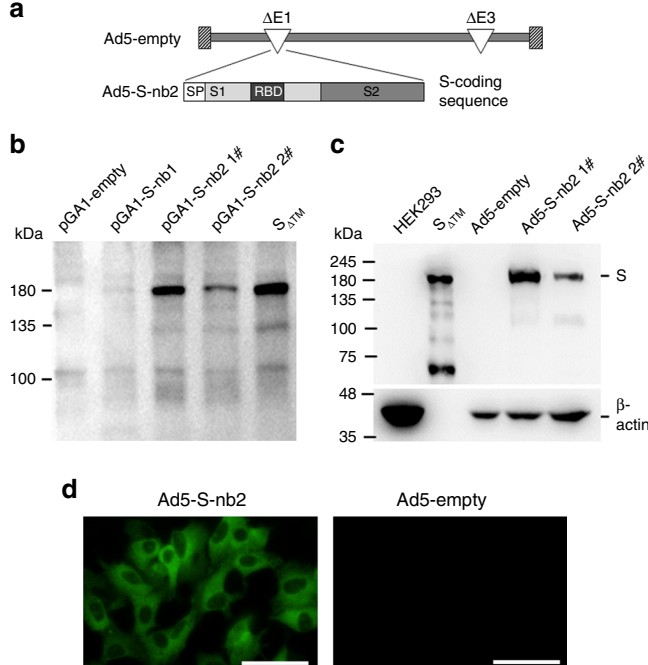

**Fig. 1 Construction and characterization of Ad5-S-nb2. a** Schematic diagram of the genome of Ad5-S-nb2 and the coding sequence for SARS-CoV-2 S protein. **b** Western blot analysis of the expression of S protein in HEK293 cells transfected with plasmids encoding an original S sequence (pGA1-S-nb1, 4 μg per well) or a codon-modified S sequence (pGA1- S-nb2 1#: 4 μg per well; pGA1- S-nb2 2#: 2 μg per well). A pGA1-empty plasmid was used as the negative control. A purified S protein with the transmembrane domain truncated (S$_{\Delta TM}$) was used as the positive control. **c** Western blot analysis of the expression of S protein in HEK293 cells infected with Ad5-S-nb2. Ad5-S-nb2 1#, 0.2 TCID50 per cell; Ad5-S-nb2 2#, 0.05 TCID50 per cell. S$_{\Delta TM}$ protein and HEK293 cells infected with Ad5-empty were examined in parallel as the positive and negative controls, respectively. The samples were derived from the same experiment and the blots were processed in parallel. **d** Immunofluorescence analysis of the expression of S protein in A549 cells mediated by Ad5-S-nb2. A549 cells were infected with Ad5-S-nb2 or Ad5-empty at 0.2 TCID50 per cell. Twenty-four hours later, cells were labeled with a human monoclonal antibody against S protein and then with an Alexa Fluor 488-conjugated mouse anti-human antibody. The cells were observed under a fluorescence microscope. Scale bar = 50 μm. For **b** and **c**, two independent experiments were carried out with similar results. For **d**, one representative result from three independent experiments is shown. Source data are provided as a Source Data file.

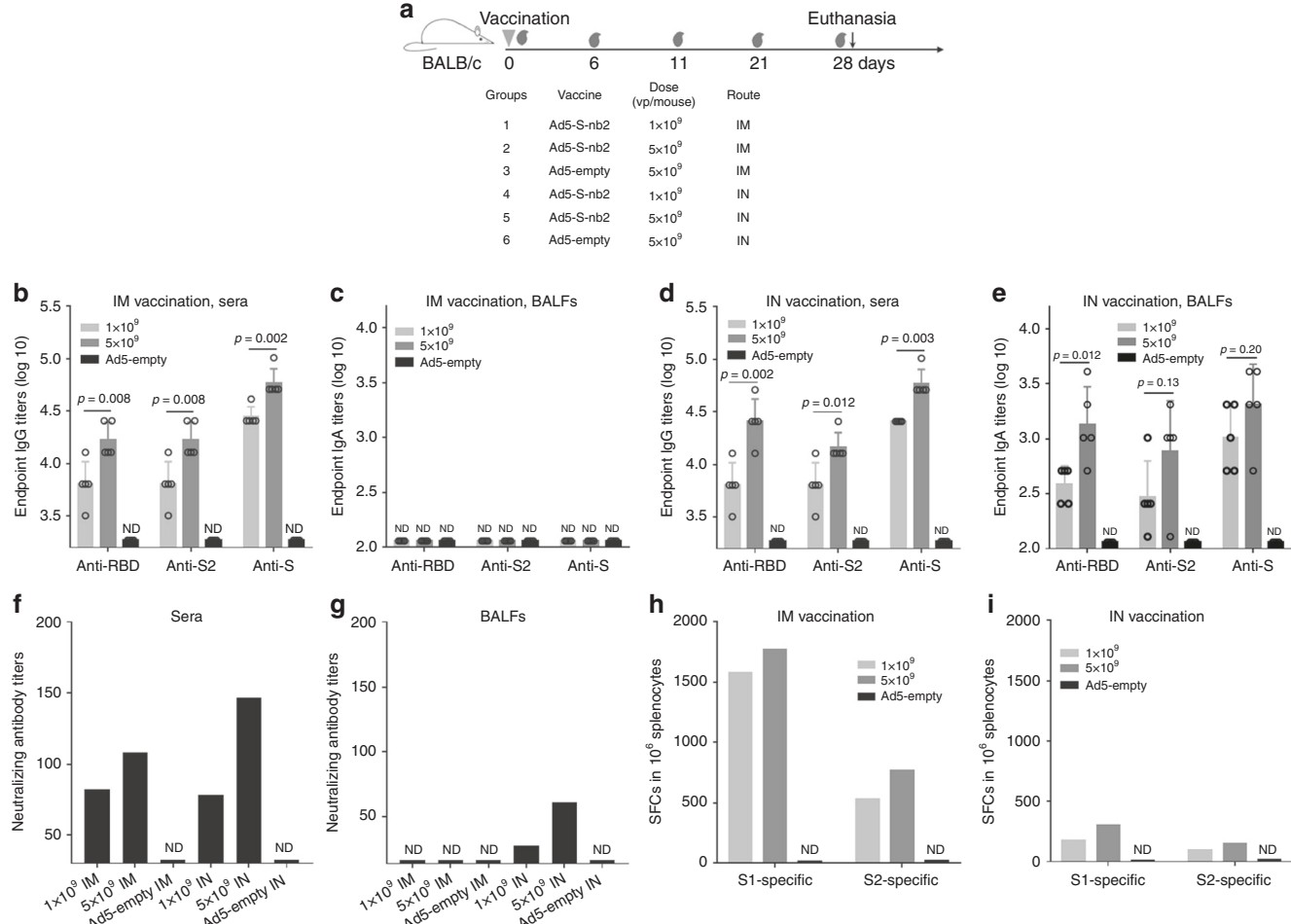

**Fig. 2 The immunogenicity of Ad5-S-nb2 in mice. a** Schematic diagram of the schedule, dosage, and delivery routes of mouse experiment. Eight-week-old BALB/c mice ($n = 5$ per group) were vaccinated with Ad5-S-nb2 using the indicated dosage through an IM or IN route. The gray blood drop symbols indicate the time points at which the serum samples were collected. Twenty-eight days after the initial vaccination, the mice were sacrificed. **b** The IgG antibody response in the sera of mice that received IM vaccination. **c** The IgA antibody response in the BALFs of mice that received IM vaccination. **d** The IgG antibody response in the sera of mice that received IN vaccination. **e** The IgA antibody response in the BALFs of mice that received IN vaccination. The endpoint titers of IgG antibodies (**b** and **d**) and IgA antibodies (**c** and **e**) against the RBD, S2, and S protein are shown. **f** The titers of neutralizing antibodies in mice sera. **g** The titers of neutralizing antibodies in mice BALFs. The sera (**f**) or BALFs (**g**) from each group of mice ($n = 5$) were pooled and assessed by S-pseudotyped reporter lentivirus. **h, j** IFN-γ-secreting cells in the spleens of mice that received IM (**h**) or IN vaccination (**j**). Splenic lymphocytes from each group of mice ($n = 5$) were pooled and stimulated with two peptide pools corresponding to the S1 and S2 region. Shown are the number of spot-forming cells (SFCs) in one million splenic lymphocytes. For **b–e**, each circle represents an individual mice, and the error bars indicate the means ± standard derivations (SD). For **f–i**, each bar reflects the mean value of three technical replicates. Data are one representative result of two independent experiments. Comparisons were performed by Student's t-test (unpaired, two-tailed). Source data are provided as a Source Data file.

On day 28 after an IN inoculation, the serum IgG responses against the S protein, RBD, and the S2 subunit increased to a similar level as IM vaccinated mice. Mice vaccinated with $5 \times 10^9$ vp had higher antibody titers than that of $1 \times 10^9$ vp, showing a dose-dependent effect (Fig. 2d). Besides, an IN inoculation of Ad5-S-nb2 elicited S-specific IgA response in the bronchoalveolar lavage fluids (BALFs) (Fig. 2e), whereas an IM injection did not give rise to significant S-specific IgA response in the BALFs (Fig. 2b). Thus, an IM injection with Ad5-S-nb2 can induce systemic IgG but not pulmonary IgA to S protein, whereas an IN inoculation can induce both systemic IgG and pulmonary IgA to S protein in mice.

We assessed the neutralizing activities in the mouse sera and the BALFs using a reporter lentivirus pseudotyped by SARS-CoV-2 S protein. Consistent to the S binding antibody response, an IM injection with $5 \times 10^9$ vp Ad5-S-nb2 elicited a higher serum neutralizing antibody response than $1 \times 10^9$ vp

(Fig. 2f). An IN inoculation with $5 \times 10^9$ vp Ad5-S-nb2 also elicited higher serum neutralizing antibody response than $1 \times 10^9$ vp (Fig. 2f). Notably, an IN inoculation dose-dependently elicited neutralizing antibody response in BALFs, whereas an IM injection with Ad5-S-nb2 elicited no significant neutralizing antibody response in BALFs (Fig. 2g). Therefore, IN vaccination can induce significant neutralizing antibodies in the pulmonary organs in mice.

To determine if Ad5-S-nb2 also elicits cell-mediated immune (CMI) response in mice, we performed an interferon-γ (IFN-γ) enzyme-linked immune spot (ELISpot) assay using lymphocytes isolated from mouse spleens. Overlapping peptides of 20 amino acids that cover S protein were synthesized and divided into the S1 and S2 pools. An IM injection with $5 \times 10^9$ vp Ad5-S-nb2 elicited a large number of spot-forming-cells (SFCs) to the S1 region but a much less response to the S2 region. $5 \times 10^9$ vp Ad5-S-nb2 elicited higher CMI response than $1 \times 10^9$ vp (Fig. 2h).

However, an IN inoculation with Ad5-S-nb2 induced much less systemic CMI response than IM vaccination in mice (Fig. 2i).

**Ad5-S-nb2 elicits antibody and CMI responses in NHPs.** We next assessed the immunogenicity of Ad5-S-nb2 in nonhuman primates (NHPs). Adult Chinese rhesus macaques (*Macaca Mulatta*) were divided into the following 5 groups: (1) An IM injection with a high dosage of $1 \times 10^{11}$ vp Ad5-S-nb2 per macaque ($n = 4$); (2) An IN and oral inoculation of $1 \times 10^{11}$ vp Ad5-S-nb2 ($n = 4$) with a half dosage administered via oral (we later found in a separate study that the oral inoculation has a negligible effect on inducing immune response in macaques so we later generally referred to this group as IN vaccination of $5 \times 10^{10}$ vp Ad5-S-nb2); (3) An IM injection with a low dosage of $1 \times 10^{10}$ vp Ad5-S-nb2 ($n = 4$); (4) An IM injection with $1 \times 10^{11}$ vp Ad5-empty as a control group ($n = 2$); and (5) Macaques without vaccination were also used as a control group ($n = 6$) (Fig. 3a). The vaccinated macaques were 6 to 14 years old, which were equivalent to about 18 to 42 years old in humans[17]. These macaques previously have been immunized with influenza viruses H1N1, H5N6, H7N7, H7N9, and an adenovirus serotype 2 vectored EBOLA vaccine in other studies (Supplementary Table 1). Thus, these macaques may be more resembling humans who have preexisting immunity to a variety of pathogens.

On day 12 after vaccination, significant serum S-specific IgG could be detected in 4 out of 4 macaques IM vaccinated with $1 \times 10^{11}$ vp Ad5-S-nb2, and in 3 out of 4 macaques IM vaccinated with $1 \times 10^{10}$ vp. Serum S-specific IgG could also be detected in 3 out of 4 IN vaccinated macaques (Fig. 3b). On day 24, serum S-specific IgG could be detected in all vaccinated macaques. The IgG titers appear to be higher in high-dose vaccinated macaques. S-specific IgG continued to increase in macaques that received either low or high dosage of IM vaccination. In contrast, S-specific IgG titers remained constant in IN vaccinated macaques after day 18. Macaques that received IN vaccination had 1–2 logs lower serum IgG titers than IM vaccinated macaques. Assessment of serum IgG responses to different regions of S protein revealed that IgG antibodies target to the whole S protein, including RBD and S2 region (Fig. 3c).

To determine if the candidate vaccine Ad5-S-nb2 also elicits CMI response in NHPs, we examined the S-specific IFN-γ-secreting cells in the peripheral blood mononuclear cells (PBMCs) to the S1 and S2 peptide pools. On day 18 after vaccination, all 8 out of 8 IM vaccinated macaques had CMI response to S1 peptide pools, whereas 6 out of 8 macaques showed CMI response to S2 peptide pools (Fig. 3d). Only 2 out of 4 IN vaccinated macaques had systemic CMI response to S1 peptide pools, and none of these macaques showed CMI response to S2 peptides. Therefore, in rhesus macaques, the CMI response mostly targets to the S1 region, similar to that observed in mice. This result suggests that IM vaccination can provoke systemic CMI response to S protein, especially to the S1 region, whereas IN vaccination elicits a weaker systemic CMI response.

**Ad5-S-nb2 protects against SARS-CoV-2 challenge in NHPs.** To assess if vaccination with Ad5-S-nb2 can confer protection against SARS-CoV-2 infection, on 30 days after vaccination, we challenged vaccinated macaques with an intratracheal inoculation of SARS-CoV-2 (strain 2019-nCoV-WIV04) at $2 \times 10^4$ 50% tissue-culture infectious doses (TCID50) (Fig. 4). Due to the space limitation of the biosafety level 4 laboratory, 3 high-dose IM vaccinated macaques and 3 IN vaccinated macaques were randomly selected for challenge (Fig. 3a). Four macaques that received no vaccination were also challenged with the same dose of SARS-CoV-2 (Fig. 3a). After challenge, all non-vaccinated macaques had virus shedding in the pharyngeal swabs for at least 10 days or throughout the experiment (Fig. 4a). Macaque C1 had virus

shedding with a peak viral load of $2.2 \times 10^6$ copies ml$^{-1}$ on day 2 to day 6 and even had $3.7 \times 10^4$ copies ml$^{-1}$ on day 10. Macaque C2 had $1.1 \times 10^4$ copies ml$^{-1}$ on day 2 and 6000 copies ml$^{-1}$ on day 10. Macaque C3 also displayed virus shedding with a peak viral load of $4.2 \times 10^6$ copies ml$^{-1}$ on day 5 and 2800 copies ml$^{-1}$ on day 7 before euthanasia. Macaque C4 had a delayed viral shedding, which appeared to be positive on day 10 (Fig. 4a). Additionally, two macaques (D1 and D2) were challenged with 400 TCID50 of SARS-CoV-2 and were also used as a control group. Macaque D1 had virus shedding throughout the study course, with a peak viral load of $2.0 \times 10^5$ copies ml$^{-1}$ on day 6 and had ~$1.0 \times 10^4$ copies ml$^{-1}$ on day 10. Macaque D2 had virus shedding throughout the study course, with a peak viral load of $1.5 \times 10^6$ copies ml$^{-1}$ on day 5 and had 570 copies ml$^{-1}$ on day 10.

The high-dose IM vaccinated macaques showed no detectable virus genome or barely above the detection limit in the pharyngeal swabs before day 5 and then became undetectable after day 7 (Fig. 4b). Macaques No. 100109 only had a viral blip (480 copies ml$^{-1}$) on day 5, and then became undetectable thereafter. Macaque No. 116004 and macaque No.116008 had viral blips on day 1 (360 and 770 copies ml$^{-1}$, respectively) and day 5 (350 and 250 copies ml$^{-1}$), and then became undetectable thereafter (Fig. 4b). Thus, an IM vaccination with $1 \times 10^{11}$ vp Ad5-S-nb2 conferred effective protection against SARS-CoV-2 infection. Interestingly, IN vaccinated macaques showed no detectable virus in the first 4 days after challenge. Macaques No.134018 and No. 140052 had a small viral bleb (~1000 copies ml$^{-1}$) around day 6 or 7 and then became undetectable thereafter. Macaque No.110113 had a viral bleb (7800 copies ml$^{-1}$) around day 6 but declined on day 7 and became undetectable on day 10 (Fig. 4c). Therefore, although IN vaccination elicited much less systemic antibody and CMI responses than IM vaccination, IN vaccination can confer effective protection against SARS-CoV-2 infection.

In light of the high-dose IM vaccination conferred effective protection, we subsequently challenged three macaques at 8 weeks after IM vaccination with a 10-folds lower dosage ($1 \times 10^{10}$ vp). Again, all three macaques were effectively protected against intratracheal SARS-CoV-2 challenge (Fig. 4d). Macaques No. 063585 and 080066 had a small viral blip (410 and 590 copies ml$^{-1}$) on day 5 but became barely detectable on day 7. Macaque No. 134056 had a small viral blip on day 3 (490 copies ml$^{-1}$) and then became undetectable thereafter.

We found no signs of antibody-dependent enhancement of infection (ADE) in all vaccinated macaques after challenge. We compared the peak viral loads in the pharyngeal swabs between vaccinated macaques and non-vaccinated macaques. The vaccinated macaques had a peak viral load of $2.9 \pm 0.4$ log10 copies ml$^{-1}$, whereas non-vaccinated macaques had a peak viral load of $5.2 \pm 1.5$ log10 copies ml$^{-1}$ (Supplementary Fig. 2). By day 10 or euthanasia, all non-vaccinated macaques were positive for virus shedding in the pharyngeal swabs, whereas all vaccinated were negative for virus shedding. Because the peak viral load does not reflect the presence of total virus over time, we next calculated the virus load based on the area under curve (AUC). The vaccinated macaques had a viral AUC of $2.7 \pm 0.6$ log10, which was 2500-folds lower than the viral AUC of $6.1 \pm 1.0$ log10 in non-vaccinated macaques (Fig. 4e). Most importantly, we assessed the presence of viral genome in biopsy samples collected from 9 anatomic locations after the macaques were euthanasia, including the trachea, the left and right bronchus, the upper, middle, and lower sites of the left and the right lung (Fig. 5a). Viral genomes were detected in the trachea ($5.7 \times 10^5$ copies ml$^{-1}$), the left bronchus ($2.4 \times 10^5$ copies ml$^{-1}$), and the lower site of the left lung ($1.8 \times 10^4$ copies ml$^{-1}$) from non-vaccinated macaque C3 on day 7. Viral genomes were also detected in the trachea

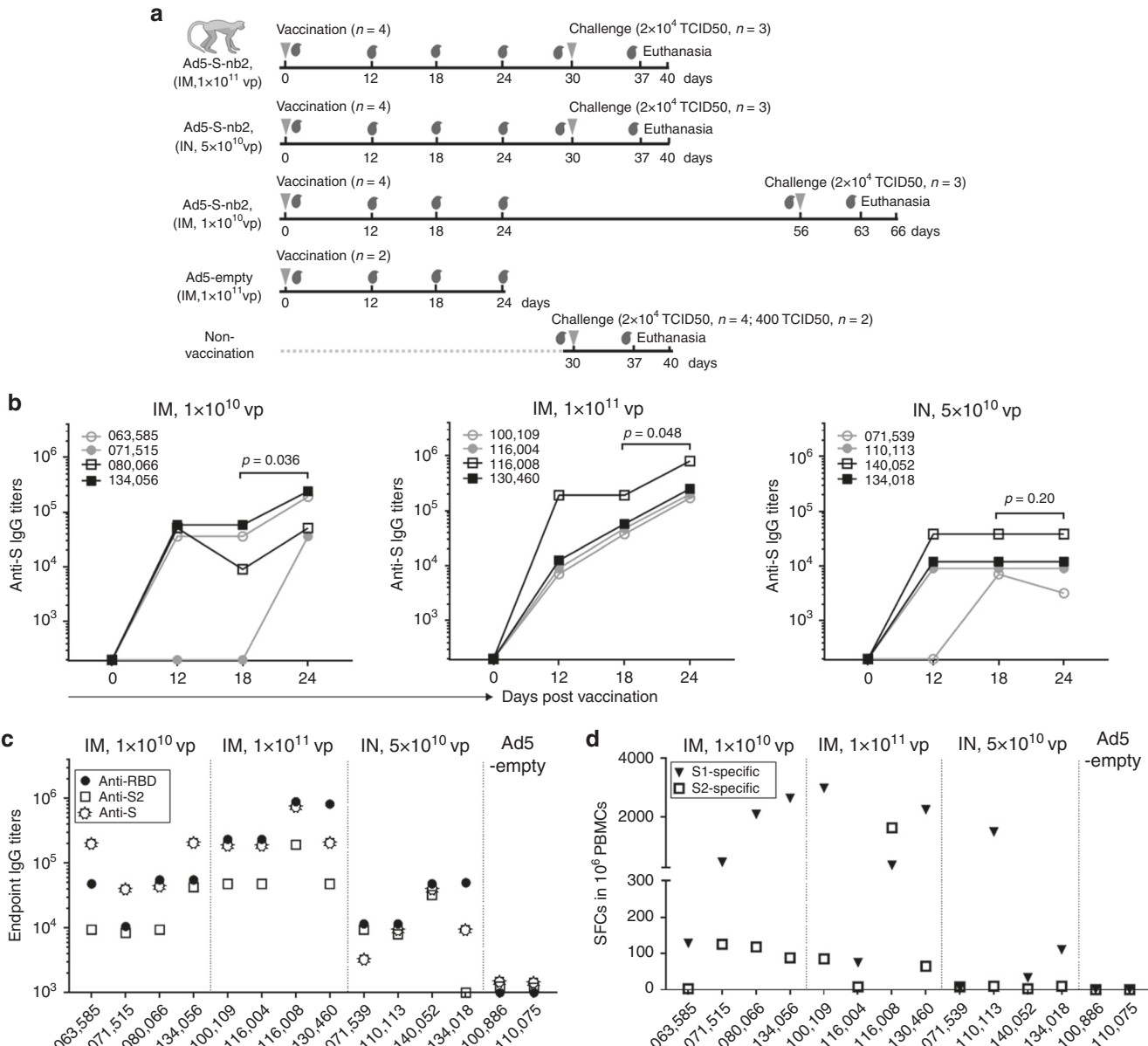

**Fig. 3 The immunogenicity of Ad5-S-nb2 in Chinese rhesus macaques. a** Schematic diagram of the vaccination and challenge studies in rhesus macaques. Three groups of macaques ($n = 4$ per group) were vaccinated via an IM injection of $1 \times 10^{10}$ vp or $1 \times 10^{11}$ vp Ad5-S-nb2 or via an IN and oral inoculation of $5 \times 10^{10}$ vp Ad5-S-nb2 each. The control groups include macaques ($n = 2$) IM injected with $1 \times 10^{11}$ vp Ad5-empty and non-vaccinated macaques ($n = 6$). On days 0, 12, 18, and 24 after vaccination, the serum samples and PBMCs were collected. On day 30 (Group 1 and 2) or 56 (Group 3) after vaccination, macaques were challenged. On day 7 or 10 after challenge, macaques were euthanatized. The gray inverted triangles indicate the time points of vaccination or challenge, whereas the gray blood drop symbols indicate the time points at which the serum samples were collected. **b** The kinetics of IgG response to the S protein in the sera. The endpoint IgG titers are shown. The overlapped data points represent the same values. Comparisons between different time points were performed by Student's $t$-test (paired, one-tailed). **c** The antigen recognition profiles of macaque immune sera. Macaque immune sera collected on day 24 were assessed for the IgG antibodies against the RBD, S2, and S protein by enzyme-linked immunosorbent assay (ELISA). **d** IFN-$\gamma$-secreting cells in the PBMCs of rhesus macaques. PBMCs isolated on day 18 were stimulated with two peptide pools corresponding to S1 and S2, respectively. Shown are the number of SFCs in one million PBMCs. All the data points represent the mean values of two technical replicates. Source data are provided as a Source Data file.

($4.7 \times 10^4$ copies ml$^{-1}$), left bronchus ($2.2 \times 10^4$ copies ml$^{-1}$) and right bronchus ($1.8 \times 10^4$ copies ml$^{-1}$) from non-vaccinated macaque C4 on day 10. In contrast, no viral genomes could be detected from all 81 biopsy samples collected from 9 vaccinated macaques, either on day 7 (4 macaques) or day 10 (5 macaques). We performed histopathological analysis on lung sections from non-vaccinated and vaccinated macaques (Fig. 5b–f). In non-vaccinated macaques, intratracheal inoculation of SARS-CoV-2 caused severe interstitial pneumonia, as evidenced by the expansion of alveolar septum, the infiltration of monocytes and lymphocytes in most alveoli, as well as the edema in a proportion of alveoli (Fig. 5b). In vaccinated macaques, the challenge caused no significant pathological abnormalities or mild histopathological changes in some macaques, which may be caused by direct administration of SARS-CoV-2 into the lungs (Fig. 5d–f).

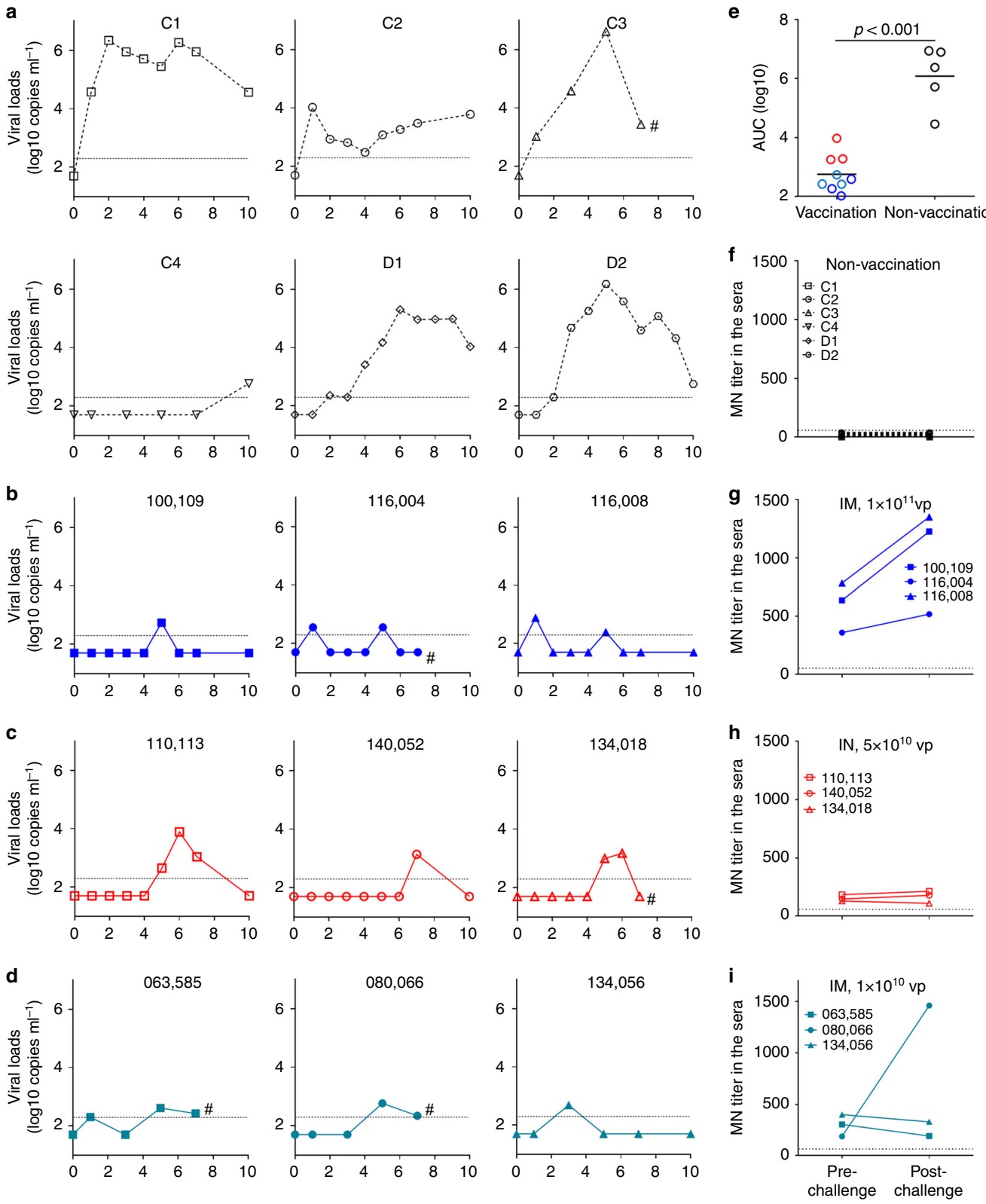

We compared the serum neutralizing antibody titers before and after the challenge by a plaque reduction neutralization test (PRNT). All non-vaccinated macaques showed no serum neutralizing activities (<1:50) against SARS-CoV-2 either before or 7 days after challenge (Fig. 4f). Before the challenge on 30 days after vaccination, three high-dose IM vaccinated macaques had serum neutralizing antibody titers with IC50 at 1:636, 1:389, and

1:784 for macaque No. 100109, No. 116004, and No.116008, respectively. On day 7 after challenge, the neutralizing antibody titers slightly increased (less than 2-folds) with IC50 at 1:1225, 1:518 and 1:1350 for macaque No. 100109, No. 116004, and No.116008 respectively, indicating no or a weak booster effect from the challenge (Fig. 4g). The three IN vaccinated macaques had lower serum neutralizing antibody titers before challenge

**Fig. 4 Ad5-S-nb2 protects against SARS-CoV-2 infection in rhesus macaques. a** The viral loads in the pharyngeal swabs of non-vaccinated macaques. **b** The viral loads in the pharyngeal swabs in macaques at 4 weeks after IM vaccination with $1 \times 10^{11}$ vp Ad5-S-nb2. **c** The viral loads in the pharyngeal swabs in macaques at 4 weeks after IN vaccination with $5 \times 10^{10}$ vp Ad5-S-nb2. **d** The viral loads in the pharyngeal swabs in macaques at 8 weeks after IM vaccination with $1 \times 10^{10}$ vp Ad5-S-nb2. The genome copy numbers in the elution of the pharyngeal swabs were determined by quantitative reverse transcription PCR (qRT-PCR). The limit of detection was 200 copies ml$^{-1}$ and marked by the dot line. The data points were expressed as the mean of two technical replicates. Macaques C3, 116004, 134018, 063585, and 080066 were euthanized on day 7 after challenge and were marked by hash symbol. Other macaques were euthanized on day 10 after challenge. Macaques C1, C2, D1 and D2 were subsequently utilized for another study. **e** The viral loads calculated based on AUC in the pharyngeal swabs in macaques after challenge. Blue circles, macaques that received $1 \times 10^{11}$ vp Ad5-S-nb2 (IM); Dark cyan circles, macaques that received $1 \times 10^{10}$ vp Ad5-S-nb2 (IM); Red circles, macaques that received mucosal vaccination (IN). Black circles, non-vaccinated macaques. Black lines reflect the mean AUC. Comparison between vaccinated ($n = 9$) and non-vaccinated ($n = 5$) macaques was conducted using Student's $t$-test (unpaired, two-tailed). **f–i** The neutralizing antibody titers against SARS-CoV-2 before and after challenge in non-vaccinated macaques (**f**), high-dose IM vaccinated macaques (**g**), IN vaccinated macaques (**h**), and low-dose IM vaccinated macaques (**i**). The sera were examined by PRNT using SARS-CoV-2 (strain 2019-nCoV-WIV04). Source data are provided as a Source Data file.

(IC50: 1:164, 1:137, and 1:150) and had no significant changes on day 7 after challenge (IC50: 1:193, 1:168, and 1:130, respectively) (Fig. 4h). For the three low-dose IM vaccinated macaques that were challenged 8 weeks after vaccination, only one macaque (No. 080066) had significant elevation of serum neutralizing antibody titer (IC50: 1:188 to 1:1460), whereas two other macaques had no elevation of serum neutralizing antibody titers (IC50: 1:315 to 1:192, and 1:400 to 1:330 for No. 063585 and No. 134056) (Fig. 4i) on day 7 post challenge. This result revealed that the inoculated virus might be eliminated immediately and no significant virus replication occurred to boost immune response in 6 out of 9 vaccinated macaques, suggesting a possibility of sterilizing immunity.

We also examined the level of anti-Ad5 neutralizing antibodies in vaccinated macaques. After vaccination, the IM vaccinated macaques had a rapid increase of serum Ad5 neutralizing antibody titers. Interestingly, the IN vaccinated macaques appeared to have an at least 6 days delayed onset of anti-Ad5 neutralizing antibodies and about 10-folds lower serum Ad5 neutralizing antibody response than the IM vaccinated macaques (Supplementary Fig. 3), indicating IN route may offer an advantage for repeated inoculation of adenovirus vectored vaccines.

## Discussion

This study demonstrated that candidate vaccine Ad5-S-nb2 can elicit S-specific antibody and CMI responses in rodents and in NHPs. A single IM injection with a low-dose of $1 \times 10^{10}$ vp Ad5-S-nb2 can confer effective protection against SAR-CoV-2 challenge in aged Chinese rhesus macaques. Non-injection vaccination routes such as an IN inoculation can also confer adequate protection. The candidate vaccine Ad5-S-nb2 confers rapid and effective virologic control as demonstrated by no detectable viral genome from each vaccinated macaque in all 9 anatomic samples collected on day 7 and day 10 after challenge.

During the review and revision of this manuscript, several candidate vaccines including a DNA vaccine and inactivated whole virus vaccines have been evaluated in rhesus macaques against SARS-CoV-2 challenge[10,18,19]. Most candidate vaccines require multiple injections. An inactivated whole virus vaccine required three injections (with alum adjuvant) to confer either partial protection (medium dose, $3 \times 3 \mu g$) or effective protection (high dose, $3 \times 6 \mu g$) at one week after the third vaccination. However, the viral genome could be detected in the lung biopsy samples from macaques vaccinated with the medium dose ($3 \times 3 \mu g$) at day 7 after challenge[18]. A DNA vaccine expressing the full-length S protein also required two injections to reduce viral replication and shedding but did not achieve sterilizing protection[10]. An Ad5 vectored COVID-19 vaccine showed good

immunogenicity in Phase I clinical trial, but the vaccination and challenge study in animal models have not been reported[11]. Of note, all reported challenge studies were done within 1 to 3 weeks after the last vaccination when the antibody response was at the highest, and there are no reports on the longevity of protective immunity. Our challenge studies were carried out at 1 month or even near 2 months after a single-dose vaccination. Interestingly, the neutralizing antibody titers did not increase in most vaccinated macaques on day 7 after challenge. This observation is different from what were observed from DNA vaccine and inactivated whole virus vaccines, which exhibited anamnestic immune response with significantly increased neutralizing antibody titers at day 7 after challenge, suggesting a booster effect by inoculated viruses and possible virus replication after challenge[10,18,19]. The absence of a booster effect in 2/3 of vaccinated macaques in our study indicated the possibility of sterilizing protection, which efficiently eliminates incoming viruses and effectively controls viral replication. Unlike inactivated whole virus vaccines that elicit only antibody response, the adenovirus vectored vaccine also elicits CMI response, which may provide an additional protection mechanism via eliminating virus-infected cells. Moreover, the IN vaccination provides mucosal immunity in the respiratory system that can effectively and immediately eliminate viruses entering the nostril and respiratory tract.

A variety of vaccine approaches have been explored for other human coronaviruses SARS-CoV and MERS-CoV in the past[13,14,20–25]. Although MERS-CoV vaccines have shown protective effects in animal models[13,14,24,25], one SARS-CoV vaccine based on modified vaccinia virus Ankara vector encoding the S protein, has been reported to cause ADE of diseases[26]. An inadequate Th1-biased T-cell response may also contribute to the immunopathology of SARS-CoV infection[27,28]. In our study, no signs of ADE were observed, indicating that Ad5-S-nb2-induced immune responses do not enhance the infection or pathogenicity of SARS-CoV-2. Consistently, other studies in rhesus macaques also showed that the immune responses elicited by a primary SARS-CoV-2 infection or inactivated whole virus effectively protected against SARS-CoV-2 challenge without observing ADE[18,19,29,30].

The safety profile of recombinant Ad5 vectored vaccines has been demonstrated in a variety of clinical trials[31]. Recently, an Ad5 vectored COVID-19 vaccine was reported to cause injection site pain, fever, and fatigue in some vaccine recipients within 24 h after vaccination, but no serious adverse events[11]. In that clinical trial, $5 \times 10^{10}$ to $1.5 \times 10^{11}$ vp Ad5 vectored COVID-19 vaccine were used, which is higher than the low-dose regimen ($1 \times 10^{10}$ vp) used in our NHP experiment. The great potency of Ad5-S-nb2 may enable the induction of protective immunity using a low dosage, and thus reduces the vaccine-associated side effects. We propose that Ad5-S-nb2 warrants further clinical evaluation as a

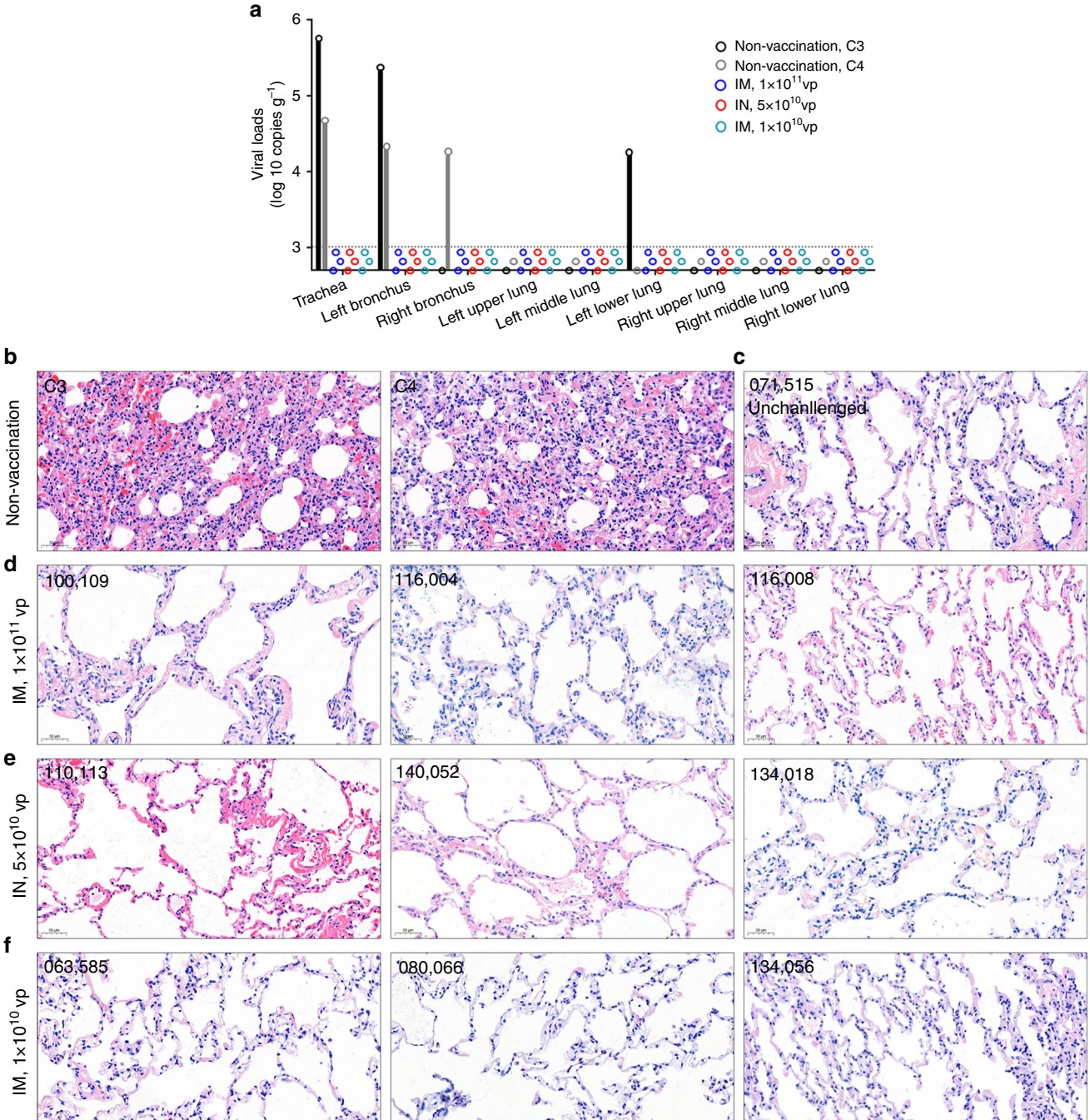

**Fig. 5 Virological and histopathological analysis of macaque airway tissues. a** The viral loads in the nine anatomic locations of macaque airway. The tissues of trachea, left and right bronchus, and the upper, middle, and lower locations of the left and right lung were collected at euthanasia. The viral genomes in the tissue homogenates were assessed by qRT-PCR. All the data points represent the mean values of two technical replicates. The dot line indicates the limit of detection (1000 copies per g tissue). **b–f** Histopathological changes in the lungs of SARS-CoC-2 challenged non-vaccinated macaques (**b**) and vaccinated macaques (**d–f**), and an unchallenged macaque IM vaccinated with $1 \times 10^{10}$ vp Ad5-S-nb2 (**c**). The lung tissue sections were stained with hematoxylin and eosin (H&E). One representative graph from each macaque is shown. Scale bars are 50 µm and indicated in each panel. Source data are provided as a Source Data file.

prophylactic vaccine for controlling the SARS-CoV-2 pandemic. Our study also demonstrated that non-injection routes such as IN administration has a great potential to confer effective protection with the ease of vaccination. The IN vaccination aimed to stimulate mucosal immunity in the respiratory tract and likely gastro-intestinal tract. Although the direct intratracheal inoculation of SARS-COV-2 in our challenge model bypassed the nostril and pharyngeal barrier which undermines the protective effect, IN

inoculation of Ad5-S-nb2 still demonstrated a great protective efficacy. The mucosal vaccination route has also been shown to induce both systemic and mucosal immunity in Ad5 vectored Ebola virus vaccines[32–34]. Mucosal immunity is important for a vaccine against respiratory viruses such as SARS-CoV-2. At present, we know little about the immune correlates of protection to SARS-CoV-2. Serum antibodies that bind to S protein and neutralizing antibodies have been mostly used for the evaluation of

convalescent plasma therapy and for vaccines. However, we found that IN vaccination induced relatively lower systemic immune responses than IM vaccination, but still conferred effective protection against SARS-CoV-2. This result implies that serum antibody titers may not necessarily be predictive of the protective efficacy. The local immune responses, i.e., the S-specific antibodies on the airway mucosa and T cell immunity, play important roles in blocking SARS-CoV-2 infection.

Ad5 prevalence in humans can reduce the efficacy of Ad5-vectored vaccines due to the preexisting anti-vector immunity from natural exposure or previous vaccination. It has been reported that preexisting Ad5 neutralizing antibodies in populations were 40%–69% in America, 77% in Southern China, and 80% in Africa[35,36]. A recent Ad5 vectored COVID-19 vaccine clinical trial indicated that a high level of Ad5 neutralizing antibodies could compromise the seroconversion of neutralizing antibodies against SARS-CoV-2 in vaccine recipients[11]. A high dosage vaccination ($1.5 \times 10^{11}$ vp) seemed to achieve a higher seroconversion rate than a low dosage ($5 \times 10^{10}$ vp). Although the preexisting Ad5 immunity may inactivate a portion of the inoculated Ad5 vectored vaccine, increasing dosage or using a more effective way to deliver Ad5 vectored vaccine may circumvent the preexisting Ad5 immunity, thus to achieve sufficient antigen-specific immune responses. It has been reported that nasal delivery of adenovirus vectored vaccines are less affected by preexisting Ad5 antibodies[33,37]. A relatively high dosage of IM injection was required for an Ad5 vectored Ebola virus vaccine to generate significant neutralizing antibodies in individuals who are seropositive for anti-Ad5 antibodies[38,39], whereas airway delivery of an Ad5 vectored vaccine circumvented the preexisting anti-vector immunity and induced protective immune response against Ebola virus infection in NHPs[33]. In our study, we found that the IN vaccinated macaques had a delayed increase and lower serum anti-Ad5 neutralizing antibody response than the IM vaccinated macaques, suggesting that IN delivery of Ad5 vectored vaccine can be a strategy for vaccination with less influence from anti-adenoviral immunity. Moreover, the administration of Ad5 vectored vaccines through nostrils may make self-vaccination possible, thus reducing the burden of healthcare workers and enabling more people to receive a vaccine within a short timeframe. In conclusion, our study demonstrated that it is possible to develop a safe and effective vaccine for controlling SARS-CoV-2 infection and preventing COVID-19. Ad5-S-nb2 is a candidate vaccine that warrants further evaluation in humans.

## Methods

**Adenovirus vectored vaccine.** Replication-incompetent recombinant adenovirus Ad5-S-nb2 was constructed as follows. In brief, the original (termed S-nb1) and codon-optimized (termed S-nb2) coding sequences for SARS-CoV-2 S protein (GeneBank ID: 43740568) were synthesized (Genescript, China). S-nb1 and S-nb2 were amplified by PCR and inserted into a shuttle plasmid pGA1 to obtain pGA1-S-nb1 and pGA1-S-nb2, respectively. The primer set for S-nb1 includes the forward primer (NB1-F: 5′-GCGTTTAAACTTAAGCTTGGTACCGAGCTCG-GATCCGCCACCATGTTTGTT TTTCTTGT-3′) and the reverse primer (NB1-R: 5′-AGAATAGGGCCCTCTAGA CTAGTTTATGTGTAATGTAATTTG-3′). The primer set for the S-nb2 includes the forward primer (NB2-F: 5′-GCGTTTAAA CTTAAGCTTGGTACCGAGCTCGGATC CGCCACCATGTTCGTGTTTCT GGT-3′) and the reverse primer (NB2-R: 5′-AGAAT AGGGCCCTCTAGACTA GTTTATCAGGTGTAGTGCAGCTTC-3′). pGA1-S-nb2 was linearized and subjected to homologous recombination with an E1 and E3 deleted pAd5ΔE1ΔE3 backbone in *E. coli* BJ5183 competent cells (Thermo Fisher). The resultant pAd5-S-nb2 was linearized and transfected into HEK293 cells (ATCC® CRL-1573™) to rescue Ad5-S-nb2. Finally, Ad5-S-nb2 and an empty Ad5 vector, Ad5-empty, were propagated, purified by cesium chloride density gradient centrifugation, titrated, and stored at −80 °C.

**SARS-CoV-2 stocks.** The SARS-CoV-2 strain 2019-nCoV-WIV04 was isolated from a COVID-19 patient in Wuhan, China (GISAID, accession no. EPI_ISL_402124). The seed SARS-CoV-2 stocks were propagated in Vero E6 cells (ATCC® CRL–1586™), which were maintained in Dulbecco's modified Eagle's medium (DMEM, Thermo Fisher) supplemented with 10% fetal bovine serum (FBS), 1 mM L-glutamine, 100 IU ml⁻¹ penicillin, and 100 μg ml⁻¹ streptomycin. At 72 h post-infection, the culture supernatants were harvested, and the virus titers were determined using a standard 50% tissue culture infection dose (TCID50) assay.

**Animal experiments.** Female BALB/c mice at 6–8 weeks of age were purchased from Beijing Vital River Laboratory Animal Technology Co. Ltd. The mice experiments were conducted in the Animal Experimental Center of GIBH, CAS. Mice were randomly allocated into 6 groups ($n = 5$ per group). The vaccination regimen was shown in Fig. 2a. Two groups of mice were vaccinated with $5 \times 10^9$ vp Ad5-empty by an IM or IN route and were used as controls. On days 0, 6, 11, 21 after vaccination, mice sera were collected. At week 4 after vaccination, mice were sacrificed, and the sera, BALFs, and splenocytes were collected and subjected to enzyme-linked immunosorbent assay (ELISA) or ELISpot assay.

A total of 20 Chinese rhesus macaques (*Macaca mulatta*, half male and female) from 6–14 years of age were included in this study. Among them, fourteen macaques involved in the vaccination were housed in the Landau Animal Experimental Center, Guangzhou. These vaccinated macaques have been previously vaccinated with influenza viruses H1N1, H5N6, H7N7, H7N9 or adenovirus serotype 2 vectored EBOLA vaccine in other studies (Supplementary Table 1). Six non-vaccinated macaques were housed in the Key Laboratory of Special Pathogens and Biosafety in Wuhan, China. The vaccination and challenge regimen was shown in Fig. 3a. Macaques were randomly allocated into 5 groups. Control group 1 ($n = 2$) was IM injected with $1 \times 10^{11}$ vp Ad5-empty, and control group 2 ($n = 6$) was non-vaccinated. The serum samples and PBMCs were collected at the indicated time points and subjected to immunological assays. The macaques were intratracheally challenged with 1 ml DMEM containing $2 \times 10^4$ or 400 TCID50 SARS-CoV-2 (strain 2019-nCoV-WIV04) in an animal biosafety level 4 laboratory (ABSL-4) in the Key Laboratory of Special Pathogens and Biosafety in Wuhan, China. Starting from the day before challenge, the pharyngeal swabs and sera were collected every day or every two days and subjected to immunological or virological assays. At day 7 after challenge, one non-vaccinated macaque (C3) and four vaccinated macaques (116004, 134018, 063585, and 080066) were euthanized. At day 10 after challenge, one non-vaccinated macaque (C4) and five vaccinated macaques (100109, 116008, 110113, 140052, and 134056) were euthanized. Biopsy samples were collected from nine anatomic locations including the trachea, the bronchus (left and right), and the lung tissues (upper, middle, and bottom, left and right). The lung tissues were subjected to hematoxylin and eosin staining. The other four non-vaccinated macaques (C1, C2, D1, and D2) were utilized for another study after day 10. All animals were anaesthetized with ketamine hydrochloride (10 mg kg⁻¹) before sample collection. The macaques were euthanized by intravenous injection of pentobarbital sodium (80 mg kg⁻¹ of body weight).

The vaccination experiments were approved by the Institutional Animal Care and Use Committee (No. 2020025 and 2020009 for mice and macaques, respectively) of GIBH. The challenge experiment in rhesus macaques was approved by the IACUC of WIV (No. WIVA21202002).

**ELISA.** The binding IgG antibodies in the immune sera were measured by ELISA. 96-well plates were coated with 0.05 μg full-length S protein, S2 or the RBD (Sino Biological Inc., China) overnight at 4 °C and blocked with 1 × phosphate-buffered saline (PBS) supplemented with 0.05% Tween-20 (PBST) and 2% bovine serum albumin (BSA) for 2 h at room temperature. The serum and BALF samples were serially diluted and added to each well. After incubation for 30 min at 37 °C, the wells were added with horseradish peroxidase (HRP)-conjugated goat anti-mouse (1:2000 in PBS containing 5% skim milk; Beyotime Biotechnology, China) or goat anti-monkey IgG antibodies (1:2000 in PBS containing 5% skim milk; Abcam). After incubation for another 30 minutes at room temperature, the reaction was developed by 3,3′,5,5′-tetramethylbenzidine (TMB) substrate and determined at 450 nm. The cutoff value was calculated as the mean optical density values at 450 nm (OD450) + 3 × standard derivations (SD) from the sera of non-vaccinated animals. The endpoint titers were calculated as the reciprocal of the highest serum dilution at which the OD450 values were equal to or greater than the cutoff value. The IgA antibodies in mouse BALFs were measured similarly using an HRP-conjugated polyclonal goat anti-mouse IgA antibody (1:2000 in PBS containing 5% skim milk; Abcam).

**ELISpot.** IFN-γ ELISpot assays were performed using freshly isolated mouse splenic lymphocytes or macaque PBMCs. In brief, sterile 96-well microtiter plates (Merck Millipore) were coated with mouse or monkey IFN-γ coating antibody (U-CyTech) at 4 °C overnight. Mouse splenic lymphocytes or macaque PBMCs were isolated using a density gradient medium (LymphoprepTM, Canada), seeded in the plates at $4 \times 10^5$ cells per well, and then stimulated with two peptide pools corresponding to the S1 region and S2 region (Genescript, China) at 2 μg ml⁻¹ per peptide. After incubation for 24 h, the plates were incubated with biotinylated detection antibodies (U-CyTech) and developed with alkaline phosphatase-conjugated streptavidin (U-CyTech) and NBT/ BCIP reagent (Pierce). Finally, the spots were counted with an ELISpot reader (Bioreader 4000, BIOSYS, Germany).

**Immunofluorescence analysis**. A549 cells (ATCC® CCL-185™) were infected with Ad5-S-nb2 or Ad5-empty at 0.2 TCID50 per cell. At 24 h post-infection, the cells were fixed with 4% paraformaldehyde. After blocking with 1× PBS containing 5% BSA, the cells were labeled with a rabbit monoclonal antibody against S protein (1:100 in PBST containing 4% BSA; Sino Biological, China) and then with an Alexa Fluor 488-conjugated goat anti-rabbit secondary antibody (1:200 in PBST containing 4% BSA; Yeasen Biotech, China). Finally, cells were observed under fluorescent microscopy.

**Micro-neutralization assay for SARS-CoV-2**. The neutralizing antibodies in Ad5-S-nb2-vaccinated rhesus macaques were measured using PRNT carried out in the ABSL-4. In brief, the serum samples were heat-inactivated at 56 °C for 90 min and serially diluted three-fold from 1:50 to1:12150. An equal volume of virus stock was mixed with the diluted sera and incubated at 37 °C for 1 h. Subsequently, 100 μl mixtures were added onto monolayer Vero cells in a 12-well plate and incubated for 1 h. The infectious mixtures were then replaced with DMEM supplemented with 2% fetal bovine serum (FBS) and 0.9% methylcellulose. Three days later, the cells were fixed with 4% formaldehyde for 30 min. Finally, the plates were washed with tap water and stained by crystal violet. The neutralizing titers were calculated as the serum dilutions at which the plaques were reduced by 50% compared to virus-only wells.

**Neutralization assay based on pseudotyped lentivirus**. In brief, lentiviral vectors pseudotyped by S protein were produced by co-transfecting the plasmid expressing SARS-CoV-2 S protein, the lentiviral vector backbone that carries an expression cassette for firefly luciferase, and the packaging plasmid into 293 T cells. The immune sera and the BALFs were serially diluted and incubated with pseudotyped virus for 1 h at 37 °C. Subsequently, the mixture was added onto a 293 T cell (ATCC® CRL-3216™) derived cell line that stably expresses human ACE2 in 96-well plates. At 48 h post-infection, the luciferase activity in the cell lysates was examined. The neutralization titers were calculated as the serum dilutions or BALF dilutions at which the luciferase activity was reduced to 50% of that from the virus-only wells.

**Neutralization assay for adenovirus**. The neutralizing antibodies against Ad5 vector in macaque sera before and after vaccination were measured as follows. In brief, the HEK293 cells were seeded into 96-well plates. One day later, serial dilutions of macaque sera were inactivated at 56 °C for 90 min and incubated with Ad5-SEAP expressing secreted-alkaline-phosphatase at $4 \times 10^6$ vp per well. The mixtures were then added to the 96-well plates and incubated for 24 h at 37 °C. Finally, the supernatants were harvested, and the SEAP activity was detected using a Phospha-Light System according to the manufacturer's instructions (Thermo Fisher). The relative light units (RLUs) were recorded and the titers were calculated as dilutions that inhibited 50% RLU values.

**qRT-PCR**. Viral RNA in the pharyngeal samples or tissue homogenates was quantified by one-step real-time quantitative RT-PCR. The swabs were placed into 1 ml DMEM. The viral RNA was extracted using a QIAamp Viral RNA Mini Kit (Qiagen). RNA was eluted in 50 μl of elution buffer and used as the template for RT-PCR. The primer set includes the forward primer (RBD-qF1: 5′-CAATGGTTTAACAGGCACAGG–3′), and the reverse primer (RBD-qR1: 5′-CTCAAGTGTCTGTGGATCACG–3′). Two μl of RNA were used to quantify the viral RNA copies by HiScript® II One Step qRT-PCR SYBR® Green Kit (Vazyme Biotech Co., Ltd) according to the manufacturer's instructions. The amplification procedures were set up as the following: 50 °C for 3 min, 95 °C for 30 s followed by 40 cycles consisting of 95 °C for 10 s, 60 °C for 30 s, and a default melting curve step in an ABI stepone machine. The standard curve was generated using serial dilutions of SARS-CoV-2 RNA fragments produced by in vitro transcription. The detection limit was determined by the standard curve and the dilution and was about 200 copies per ml swab elutes or about 1000 copies per g biopsy samples. The viral loads were calculated as the genome copies of SARS-CoV-2 in one ml swab elutes or in one g tissues. The area under curve was calculated using the Graphpad Prism version 7.0.

**Western blot analysis**. HEK293 cells in 6-well plates were transfected with pGA1-NB1 (4 μg per well), pGA1-NB2 (4 μg or 2 μg per well) or pGA1-empty (4 μg per well), or infected with Ad5-S-nb2 or Ad5-empty at 0.2 or 0.05 TCID50 per cell. At 24 h after transfection or infection, the cells were harvested, treated with lysis buffer, and subjected to SDS-PAGE. After transfer, the membranes were incubated with a rabbit monoclonal antibody against SARS-CoV-2 S protein (Sino Biological, China) and then were incubated with HRP-conjugated goat anti-rabbit secondary antibodies (SeraCare Life Sciences). Finally, the membranes were developed with a chemiluminescent HRP substrate (Merck). The expression of β-actin was also examined in parallel as an internal control.

**Quantification and statistical analyses**. All quantifications were performed unblinded. Statistical parameters including the definitions and exact value of $n$ (e.g., total number of animals and replications), deviations, $p$ values, and the types of the statistical tests are reported in the figures and the corresponding legends.

Analysis of virological and immunological data was performed using the GraphPad Prism version 7.0 (GraphPad Software). Comparisons between groups were conducted using unpaired Students' $t$-test (two-tailed). Comparisons between different time points in the same group were conducted using paired Students' $t$-test (one-tailed). Differences were considered statistically significant when $p$ values were less than 0.05. Data graphs were constructed using GraphPad Prism version 7. Figures and illustrations were created using Photoshop version CS5 (Adobe Systems Inc.) and Microsoft Powerpoint version 2010 (Microsoft).

**Reporting summary**. Further information on research design is available in the Nature Research Reporting Summary linked to this article.

## Data availability
The data that support the findings of this study are available from the corresponding authors upon reasonable request. Source data are provided with this paper.

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

## Acknowledgements

We are particularly grateful to the running team of the Key Laboratory of Special Pathogens and Biosafety, Wuhan Institute of Virology for their assistance in testing vaccine efficacy. We thank the staff at the Animal Center of GIBH and Guangdong Landau Biotechnology Co. Ltd. for their excellent technical assistance. This work was partly supported by the National Natural Science Foundation of China for SARS-CoV-2 (82041014), the Strategic Priority Research Program of the Chinese Academy of Sciences (XDB29050701), China Evergrande Group funding for SARS-CoV-2 (2020GIRHHMS22), and Guangzhou Health Care and Cooperative Innovation Major Project.

## Author contributions

L.F., Z.Y. and L.C. designed the studies. S.G., C.Y., X.L. and Y.X. developed the Ad5 vectored vaccines. Q.W., C.Y., Y.L., B.L., X.Z. and Y.F. conducted the Western blot analysis and immunological assays. Q.W., P.H., F.Z., P.L. and X.L. conducted mouse studies. Q.W., Y.L., Y.G. and J.L. conducted rhesus macaque studies. C.S., J.W., Y.L., R.J. and Y.Z conducted the challenge assays. J.W., J.S. and C.K. conducted the virological assays. X.N., C.L., W.P., J.Z., X.C., T.X. and N.Z. reviewed the manuscript. L.F., Z.Y. and L.C. wrote the paper with all co-authors.

## Competing interests

S.G., C.Y., Y.X., B.L. and X.L. are employees of Guangzhou nBiomed Ltd. L.C. serves as Chief Scientific Advisor for Guangzhou nBiomed Ltd. The remaining authors declare no competing interests.
