## [Peer Review File · Nature Communications]

Reviewers' Comments:

Reviewer #1:

Remarks to the Author:

This paper describes immune responses provided by a recombinant adenovirus displaying the S protein of SAR-CoV-2 in mice and macaques. Although other research groups are adopting the same strategy to find a vaccine against COVID-19 it is of interest to have contrasting results from different experiments performed in different conditions by different researchers. In that respect the article is not original but given the critical situation caused by this pandemic virus, results obtained in this study will be of major interest for a wide scientific community. The most important part of the work is centered in comparing diverse routes and doses of immunization in NHPs. Upon challenge, 30 days after a single immunization, protection seems to be effective and results are quite promising. However, and based on only 2 control animals (non-immunized) it is difficult to appreciate the severity of the challenge. Indeed, these animals show very different profiles of infection. Since SARS-CoV-2 does not provoke an acute pathology in macaques (as stated by the authors) it is questionable if the vaccine will be sufficiently potent in humans. It is a critical point to add in the discussion. Nonetheless the vaccine provides a notable reduction of virus replication. Also, there is numerous grammatical errors and statistics are difficult to make with such reduce number of macaques. Below is a non exhaustive list of corrections and questions for the authors.

L48: replace "have not been available" by are not available.

L55: remove "such as" and replace by angiotensin

L57: article missing: The S

L58: for instead of "of"

L82 and in many places of the manuscript: the S, the RBD, the S2 etc...

L172: "significant" statistics have been done?

L194-195: what is a challenge provoked by a natural infection? it has been quantified? transmission experiments should be conducted to assess the veracity of such statement.

L199: replace "strong" by some protective efficacy

L212: this speculation is not very well funded and should be removed

L221: for MERS-CoV, vaccines based on the S protein have shown efficacy and protection in target animals (camelids) which is more relevant than animal models with the exception of NHPs, cite them.

L227: antibodies at the mucosal surface have not been measured in this study in macaques. Please correct.

L244: replace "general" by global.

Concerning the references, reference number 10 of Wang et al, is outdated (2015) replace by a more recent review.

In the extended data table 2: mention the animal species.

Reviewer #2:

Remarks to the Author:

Reviewer comments for Manuscript: An adenovirus vectored COVID-19 vaccine conferred effective protection against SARS-COV-2 challenge in rhesus macaques

This manuscript describes the production of a COVID-19 vaccine that is based on the use of an E1/E3 deleted recombinant Ad5 vectored vaccine expressing a codon-optimized SARS-CoV2 Spike protein. The vaccine was tested in mice and macaques and found to induce detectable anti-CoV2 antibodies and T cells. Upon challenge, the macaques had significant reductions in viral loads as compared to the controls. There are a few major concerns in the experimental design of this study that may significantly reduce the impact and value of the data.

- 1) The animal group numbers are very small. This is understandable for a macaque study, however, a control group of 2 animals is unacceptable. Especially given that there was ≥ 1000 -fold differences in viral loads between animals. In many ways the consistency and reproducibility of the control group is essential to the rest of the study. Depending on the order of the challenge procedures it is possible that there was a degradation of the challenge virus during the procedure where the first challenged received a higher infectious dose than the last. A TCID50 assay of the starting material and after the final challenge would rule out this possibility.
- 2) With such small groups statistical analyses would be under powered and subject to stochastic error.
- 3) There were no experimental repeats. A dose-dependent protective response in the challenge macaques would have added much to the study. Why was the low dose I.M. group excluded from the challenge.
- 4) The mouse data and the macaque I.N. neutralization data does not correlate. Typically I.M. immunization with rAd results in higher neutralizing antibodies as compared to the I.N. route as is seen in the macaques. However, the mice showed the opposite, where the I.N. mice had higher systemic sera neutralizing antibodies than the I.M. immunized mice using the high dose.

Other minor concerns:

- 1) There are no defining correlates of immunity established for a CoV2 vaccine however, there are described neutralizing titers in convalescent sera. It would be good to include some discussion on the Neutralizing titers of patients that have recovered from COVID-19 or that is used for convalescent sera therapy.
- 2) There is a recent Lancet publication on Ad5-CoV2 vaccine in humans. It would be good to include some discussion on the Neut antibody titers in that study, especially since it included analyses on vaccinees with high and low anti-Ad5 neut antibodies.
- 3) General English language issues.
- 4) Why is the vaccine referred to as NB2?
- 5) Line 126, change to 5×10^{10} vp for consistency.
- 6) The male in the control was small for the age. The other 9 year old males were significantly bigger. Was there any issues with this macaque. This reiterates the importance of the control group.
- 7) Is there any reason why the macaques would have so much variation in the ELISPOT assay?
- 8) Why would there be less of an impact of pre-existing immunity on I.N. immunization as compared to I.M.? Especially since this is a respiratory virus and even the mouse data suggests that I.N. immunization induces higher levels of Neut antibody.
- 9) Fig. 2 g and h, the low dose bar graph is too light to see when the document is printed.

Reviewer #3:

Remarks to the Author:

In this study, Feng and colleagues developed a replication defective Ad5 expressing full length spike of SARS-CoV-2 and vaccinated mice and Chinese rhesus macaques. Intramuscular and intranasal immunizations were tested. 50% of the vaccinated rhesus macaques (n=6) were challenged along with 2 empty vector vaccinated macaques to evaluate protection. The binding antibody data in both species is clear and conclusively show the presence of anti-spike antibody targeting RBD, S2 and S. The neutralization data and protection results are not conclusive for the reasons mentioned below. Overall, based on these data one can conclude that a single Ad5 vaccination is strongly immunogenic to induce binding antibody and T cell responses but the protective effects of the vaccine are preliminary and inconclusive.

Lines 74-75 – can the authors add these data? If not, clearly state data not shown.

Fig 3D, Lines 146-148: A neutralizing titer of 10 or below is considered positive. This is not justified/validated. To make this conclusion, the authors need to show titers prevaccination and in a group of unvaccinated animals to establish a cut-off. Without these data the positivity for neutralization is questionable.

There are multiple issues with interpretation of challenge outcome. First, the challenge was performed only via intratracheal route and intranasal infection was not included. This is important to mimic human transmission. Second, only two controls were included and only one of them exhibited strong virus replication in the nasopharyngeal secretions. Third, virus replication was not measured in the BAL where challenge occurred. Fourth, it is not clear if there was any pathology in the unvaccinated animals. A comment was made only for the vaccinated animals. These data should be presented. Fifth, six out of 12 vaccinated animals were selected for challenge and it is not clear on what basis these were selected? Because of this selection it is hard to interpret the protection outcome. Overall, these results are very preliminary.

Two kinds of microneutralization assays were used for monkey sera. One based on cytopathic effects (Fig. 3D, Day 18 post vaccination, low or no titer) and the other based on plaque reduction titer (Fig. 4E, Day 30 post vaccination, titers between 100-700). They seem to give drastically different results. This again raises the validity of these assays. It is very important to show the titers for each assay prior to vaccination and potentially against in the human convalescent sera. It is also important to note that pseudovirus neutralization assay was used for mouse sera which is at least 10-times more sensitive than actual virus neutralization.

The prevaccination anti-Ad5 neutralization titer is two logs higher in the empty vector vaccinated animals compared to Spike vaccinated animals. This could have limited the Ad5 replication in the controls and accordingly innate activation. This is an important point considering an association between BCG vaccination and protection in humans that was presumed to be due to innate activation induced by BCG.

Reviewer #1 (Remarks to the Author):

This paper describes immune responses provided by a recombinant adenovirus displaying the S protein of SAR-CoV-2 in mice and macaques. Although other research groups are adopting the same strategy to find a vaccine against COVID-19 it is of interest to have contrasting results from different experiments performed in different conditions by different researchers. In that respect the article is not original but given the critical situation caused by this pandemic virus, results obtained in this study will be of major interest for a wide scientific community. The most important part of the work is centered in comparing diverse routes and doses of immunization in NHPs. Upon challenge, 30 days after a single immunization, protection seems to be effective and results are quite promising. However, and based on only 2 control animals (non-immunized) it is difficult to appreciate the severity of the challenge. Indeed, these animals show very different profiles of infection. Since SARS-CoV-2 does not provoke an acute pathology in macaques (as stated by the authors) it is questionable if the vaccine will be sufficiently potent in humans. It is a critical point to add in the discussion. Nonetheless the vaccine provides a notable reduction of virus replication.

Response: We appreciate your positive comments and suggestions. In the revised manuscript, we added four more non-vaccinated macaques as controls (lines 180-193). We also added another three macaques for a challenge at 8 weeks after vaccination with a single low-dose (1×10^{10} vp) (lines 210-216). Therefore, a total of 9 vaccinated macaques and 6 non-vaccinated macaques were used for SARS-CoV-2 challenge

study. We have added histopathological data in the revised manuscript (Figure 5).

Also, there is numerous grammatical errors and statistics are difficult to make with such reduce number of macaques. Below is a non exhaustive list of corrections and questions for the authors.

L48: replace "have not been available" by are not available.

L55: remove "such as" and replace by angiotensin.

L57: article missing: The S

L58: for instead of "of"

L82 and in many places of the manuscript: the S, the RBD, the S2 etc...

L172: "significant" statistics have been done?

L194-195: what is a challenge provoked by a natural infection? It has been quantified transmission experiments should be conducted to assess the veracity of such statement.

L199: replace "strong" by some protective efficacy

L212: this speculation is not very well funded and should be removed

L221: for MERS-CoV, vaccines based on the S protein have shown efficacy and protection in target animals (camelids) which is more relevant than animal models with the exception of NHPs, cite them.

L227: antibodies at the mucosal surface have not been measured in this study in macaques. Please correct.

L244: replace "general" by global.

In the extended data table 2: mention the animal species.

Response: Many thanks for your suggestions. We have made revisions according to your suggestions.

Reviewer #2 (Remarks to the Author):

This manuscript describes the production of a COVID-19 vaccine that is based on the use of an E1/E3 deleted recombinant Ad5 vectored vaccine expressing a codon-optimized SARS-CoV2 Spike protein. The vaccine was tested in mice and macaques and found to induce detectable anti-CoV2 antibodies and T cells. Upon challenge, the macaques had significant reductions in viral loads as compared to the controls. There are a few major concerns in the experimental design of this study that may significantly reduce the impact and value of the data.

Please see the following.

1) The animal group numbers are very small. This is understandable for a macaque study, however, a control group of 2 animals is unacceptable. Especially given that there was ≥ 1000 -fold differences in viral loads between animals. In many ways the consistency and reproducibility of the control group is essential to the rest of the study. Depending on the order of the challenge procedures it is possible that there was a degradation of the challenge virus during the procedure where the first challenged received a higher infectious dose than the last. A TCID50 assay of the starting material and after the final challenge would rule out this possibility.

Response: We appreciate your comments and suggestions. We have added four more non-vaccinated macaques as controls (lines 180-193). We also added another three macaques for SARS-CoV-2 challenge at 8 weeks after vaccination with a lower dosage (1×10^{10} vp) (lines 210-216). Therefore, a total of 9 vaccinated macaques and 6 non-vaccinated macaques were used for SARS-CoV-2 challenge study.

We also noticed that the variation of the viral loads in the two macaques in the control group, so now we have added 4 more macaques in the control group. Over 1000-fold differences in viral loads between animals were also observed in a recent report using India rhesus macaques challenged with SARS-CoV-2 (A. Chandrashekar et al., Science 2020, doi:10.1126/science.abc4776). Another study using Chinese rhesus macaques challenged by 1×10^6 TCID50 SARS-CoV-2 also showed a nearly 100-fold difference in the viral loads in throat swabs (Q. Gao et al., Science 2020, doi:10.1126/science.abc1932). We believe that this variation is most likely due to the individual difference of macaques. It is unlikely attributed to the degradation of the challenge viral stocks because: 1) we tested the stability of the viral stocks and found that the titers remain unchanged for up to 7 days at 4°C; and 2) we also examined the viral titers in the stocks before and after challenge, and no difference was detected.

2) With such small groups statistical analyses would be under powered and subject to stochastic error.

Response: We have added four more non-vaccinated macaques as controls. We also added another three macaques for SARS-CoV-2 challenge at 8 weeks after vaccination with a lower dosage (1×10^{10} vp). Therefore, a total of 9 vaccinated macaques and 6 non-vaccinated macaques were used for evaluation of SARS-CoV-2 challenge study.

3) There were no experimental repeats. A dose-dependent protective response in the

challenge macaques would have added much to the study. Why was the low dose I.M. group excluded from the challenge.

Response: Due to the space limitation of biosafety level 4 laboratory, we could not include all vaccinated macaques in one challenge study, so three high-dose IM vaccinated macaques and three IN vaccinated macaques were randomly selected in the first challenge study. We have performed another challenge study for macaques at 8 weeks after a low-dose IM vaccination. Again, all macaques were protected, which not only repeated and conformed the initial finding but also demonstrated that even a lower vaccine dosage could confer protection at 8 weeks after vaccination (Other reported challenge studies were done at 1-3 weeks after a booster vaccination, when the immune response was at the peak). The results have been added to the revised manuscript (lines 210-216).

4) The mouse data and the macaque I.N. neutralization data does not correlate. Typically I.M. immunization with rAd results in higher neutralizing antibodies as compared to the I.N. route as is seen in the macaques. However, the mice showed the opposite, where the I.N. mice had higher systemic sera neutralizing antibodies than the I.M. immunized mice using the high dose.

Response: We noticed that in the high dose group (5×10^9 vp) but not the low dose group (1×10^9 vp), the IN vaccination resulted in 40% higher serum neutralizing antibodies than IM immunization in mice. We think that the high dose (5×10^9 vp) may have reached saturation for an IM injection, but not for an IN inoculation in mice. The

nasopharyngeal cavity and airway may provide more surface area than the locally injected muscle site. An IN inoculation may allow Ad5-S-nb2 to infect more airway epithelium cells, and thus more S proteins can be expressed to induce a greater antibody response in mice. Our result also coincided with a recent study which also showed IN route is better than IM route for an Ad5 vectored MERS-CoV S vaccine to provoke S-specific binding and neutralizing antibodies in mouse sera (Kim MH et al., PLoS ONE 2019, doi: 10.1371/journal.pone.0220196). Apart from the mouse study, we consider the use of rhesus macaques as an NHP model is more relevant to humans.

Other minor concerns:

- 1) There are no defining correlates of immunity established for a CoV2 vaccine however, there are described neutralizing titers in convalescent sera. It would be good to include some discussion on the Neutralizing titers of patients that have recovered from COVID-19 or that is used for convalescent sera therapy.

Response: We have added this into the discussion in the revised manuscript (lines 340-342).

- 2) There is a recent Lancet publication on Ad5-CoV2 vaccine in humans. It would be good to include some discussion on the Neut antibody titers in that study, especially since it included analyses on vaccinees with high and low anti-Ad5 neut antibodies.

Response: We appreciate your suggestion. We have added a sentence in the revised

manuscript to discuss this issue (lines 353-357).

3) General English language issues.

Response: We have invited a native English speaker to revise our manuscript.

4) Why is the vaccine referred to as NB2?

Response: We designed a series of S protein variants, including codon-optimization, deletion of transmembrane motifs, amino acid (proline) substitution, elimination of protease sites, and signal peptide replacement. We named these variant designs from NB1-NB12, among which NB2 showed efficient expression in human cells and stimulated best antibody response. Therefore, gene NB2 was selected for inserting into Ad5 vector. To avoid confusion, we now renamed this Ad5 vector as Ad5-S-nb2 in the revised manuscript.

5) Line 126, change to 5×10^{10} vp for consistency.

Response: We have changed 0.5×10^{11} to 5×10^{10} vp for consistency in the revised manuscript.

6) The male in the control was small for the age. The other 9 year old males were significantly bigger. Was there any issues with this macaque? This reiterates the importance of the control group.

Response: This macaque was used as a control for a reason it was available. In the

revised manuscript, we added four more non-vaccinated macaques as controls. We also added another three macaques for SARS-CoV-2 challenge at 8 weeks after vaccination with a lower dosage (1×10^{10} vp). Therefore, a total of 9 vaccinated macaques and 6 non-vaccinated macaques were used for SARS-CoV-2 challenge study.

7) Is there any reason why the macaques would have so much variation in the ELISPOT assay?

Response: We speculated that the variation in the ELISpot assay was attributed to the individual difference among rhesus macaques. A DNA based COVID-19 vaccine in macaques also showed a considerable variation (J. Yu et al., Science 2020, doi: 10.1126/science.abc6284). The spot-forming-cells ranged from non-detectable to near 1000 per one million PBMCs, similar to that observed in our study.

8) Why would there be less of an impact of pre-existing immunity on I.N. immunization as compared to I.M.? Especially since this is a respiratory virus and even the mouse data suggests that I.N. immunization induces higher levels of Neut antibody.

Response: Several studies demonstrate that an IN inoculation with Ad5 vectored vaccine is less affected by the pre-existing anti-Ad5 immunity (J. Richardson et al., J Infec Dis 2011, doi: 10.1093/infdis/jir332; J. Richardson et al., J Viral 2013, doi: 10.1128/JVI.02864-12; M. Croyle et al., PLOS One 2008, doi:

10.1371/journal.pone.0003548; Z. Xiang et al., J Virol 2003, doi: 10.1128/JVI.77.20.10780-10789.2003). Possible explanations include: 1) In a natural infection, the amount of virus is relatively small and can be blocked by a small number of antibodies present in the local mucus. However, inoculation of a large quantity of Ad5 vectors (1×10^{10} to 1×10^{11} vp) could mostly escape the antibody neutralization, as the number of vaccine vectors outweighs the number of neutralizing antibodies present in nasopharyngeal tract; 2) The airway lumen and oral cavity have large surface areas that are covered by epithelial cells expressing Coxsackie and Adenovirus receptor (CAR) which is used by Ad5 to infect target cells. The presence of a large amount of susceptible cells for Ad5 infection can overcome neutralization by anti-Ad5 antibodies (Z. Xiang et al., J Virol 2003, doi: 10.1128/JVI.77.20.10780-10789.2003).

9) Fig. 2 g and h, the low dose bar graph is too light to see when the document is printed.

Response: Thanks for your kind suggestion. We have replaced the original Fig. 2g.

Reviewer #3 (Remarks to the Author):

In this study, Feng and colleagues developed a replication defective Ad5 expressing full length spike of SARS-CoV-2 and vaccinated mice and Chinese rhesus macaques. Intramuscular and intranasal immunizations were tested. 50% of the vaccinated rhesus macaques (n=6) were challenged along with 2 empty vector vaccinated macaques to evaluate protection. The binding antibody data in both species is clear and conclusively show the presence of anti-spike antibody targeting RBD, S2 and S. The neutralization data and protection results are not conclusive for the reasons mentioned below. Overall, based on these data one can conclude that a single Ad5 vaccination is strongly immunogenic to induce binding antibody and T cell responses but the protective effects of the vaccine are preliminary and inconclusive.

1. Lines 74-75 – can the authors add these data? If not, clearly state data not shown.

Response: Thanks for your suggestion. To avoid confusion, we deleted this description in the revised manuscript.

2. Fig 3D, Lines 146-148: A neutralizing titer of 10 or below is considered positive.

This is not justified/validated. To make this conclusion, the authors need to show titers prevaccination and in a group of unvaccinated animals to establish a cut-off.

Without these data the positivity for neutralization is questionable.

Response: We agree with your concern. In our initial submitted manuscript, we used two different methods that were performed by two different labs using different virus strains to measure neutralizing activities of serum samples. To avoid confusion, we

later only used plaque reduction titer method to measure more samples, including non-vaccinated macaques. Please see Figure 4 for neutralizing antibody results.

3. There are multiple issues with interpretation of challenge outcome. First, the challenge was performed only via intratracheal route and intranasal infection was not included. This is important to mimic human transmission. Second, only two controls were included and only one of them exhibited strong virus replication in the nasopharyngeal secretions. Third, virus replication was not measured in the BAL where challenge occurred. Fourth, it is not clear if there was any pathology in the unvaccinated animals. A comment was made only for the vaccinated animals. These data should be presented. Fifth, six out of 12 vaccinated animals were selected for challenge and it is not clear on what basis these were selected? Because of this selection it is hard to interpret the protection outcome. Overall, these results are very preliminary.

Response:

1) We agree with you that an intranasal challenge would more closely mimic human transmission than an intratracheal challenge. However, so far, there is no well-established intranasal challenge model to ensure that all macaques would be infected. An intratracheal challenge can ensure all challenged macaques to be infected. This is important because the number of macaques is small in each group. Intratracheal challenge with as little as 400 TCID₅₀ could result in infection in rhesus macaques. In fact, we were concerned that intratracheal challenge with 2×10^4 TCID₅₀

may undermine mucosal immunity mediated protective efficacy because it bypassed the nasopharyngeal barrier. Nevertheless, we still observed effective protection in IN vaccinated macaques, demonstrating the potency of IN vaccination in establishing mucosal immunity in the respiratory tract.

2) In the revised manuscript, we have added four more non-vaccinated macaques as controls (lines 180-193). We also added another three macaques for challenge at 8 weeks after IM vaccination with a lower dosage (1×10^{10} vp) (lines 210-216). Therefore, a total of 9 vaccinated macaques and 6 non-vaccinated macaques were used for evaluation of SARS-CoV-2 challenge study.

3) We did not perform bronchoalveolar lavage to measure viral load. Alternatively, we measured the viral loads in nine biopsy samples collected from the trachea, bronchus (left and right), and lung (upper, middle, bottom, left and right). We believe the detection of 9 different anatomic samples in the pulmonary organs can provide reliable data.

4) We have obtained the pathology data and have added in the revised manuscript. Please see Figure 5. In non-vaccinated macaques, SARS-CoV-2 caused severe interstitial pneumonia, as evidenced by the expansion of alveolar septae, the infiltration of monocytes and lymphocytes in most alveoli, as well as edema in a proportion of alveoli. In vaccinated macaques, we observed no significant pathological abnormalities caused by SARS-CoV-2 or only mild histopathological changes that may be caused by direct intratracheal inoculation of the viruses into the lungs.

5) Due to the space limitation of biosafety level 4 laboratory, we could not include all vaccinated macaques in one challenge study, so three high-dose IM vaccinated macaques and three IN vaccinated macaques were randomly selected in the first challenge study. We subsequently performed another challenge study for macaques at 8 weeks after a low-dose IM vaccination. Again, all macaques were protected, which not only repeated and conformed the initial finding but also demonstrated that even a 1/10 lower vaccine dosage could confer protection long after vaccination (Other reported challenge studies were done within 1-3 weeks after a booster vaccination, when the antibody response was at the highest peak). The results have been added in the revised manuscript (lines 210-216).

4. Two kinds of microneutralization assays were used for monkey sera. One based on cytopathic effects (Fig. 3D, Day 18 post vaccination, low or no titer) and the other based on plaque reduction titer (Fig. 4E, Day 30 post vaccination, titers between 100-700). They seem to give drastically different results. This again raises the validity of these assays. It is very important to show the titers for each assay prior to vaccination and potentially against in the human convalescent sera. It is also important to note that pseudovirus neutralization assay was used for mouse sera which is at least 10-times more sensitive than actual virus neutralization.

Response: We agree with your concern. In our initial submitted manuscript, we used two different methods that were performed by two different labs (in Guangzhou and

Wuhan) using different virus strains to measure neutralizing activities. To avoid confusion, we subsequently used one method (plaque reduction method) to measure more samples. We only measured the neutralizing activities of mouse sera using pseudotyped lentivirus because of the limited access to use real SARS-CoV-2 in the BSL-4 laboratory. Mice were used to demonstrate the immunogenicity of our candidate vaccine before we move forward the study in NHPs. We consider rhesus macaque is a more human relevant model than mouse.

5. The prevaccination anti-Ad5 neutralization titer is two logs higher in the empty vector vaccinated animals compared to Spike vaccinated animals. This could have limited the Ad5 replication in the controls and accordingly innate activation. This is an important point considering an association between BCG vaccination and protection in humans that was presumed to be due to innate activation induced by BCG.

Response: Please note that Ad5-S-nb2 is replication-incompetent as it has a deletion of E1 gene and can only replicate in HEK293 cells that provides E1 gene products *in trans*. Two Ad5-empty injected macaques (No 100886 and 110075) were reused from previous experiments for other Ad5 vectored vaccines, so the baseline Ad5 neutralizing antibody titers were higher. These macaques were used to show that no S binding antibodies were generated by injection of Ad5-empty. We have added the background information for all these macaques in the revised Supplementary Table 1. The replication-incompetent adenovirus on innate activation is transient that has no

effect on our challenge protection because the challenges were performed on up to 2 months post-vaccination. In contrast, BCG is mycobacterium that can replicate and colonize in infected hosts and continuously stimulate the innate immune system.

Reviewers' Comments:

Reviewer #2:

Remarks to the Author:

I am satisfied with the revisions. The manuscript has been significantly improved.

Reviewer #3:

Remarks to the Author:

The authors have addressed my concerns satisfactorily.